# An Information Entropy Masked Vision Transformer (IEM-ViT) Model for Recognition of Tea Diseases

**Jiahong Zhang [1,2], Honglie Guo [1,3], Jin Guo [4] and Jing Zhang [1,3,*]**

1    Faculty of Information Engineering and Automation, Kunming University of Science and Technology, Kunming 650500, China
2    Yunnan Key Laboratory of Green Energy, Electric Power Measurement Digitalization, Control and Protection, Kunming 650500, China
3    Yunnan Key Laboratory of Computer Technology Applications, Kunming 650500, China
4    Faculty of Big Data, Yunnan Agricultural University, Kunming 650051, China
*    Correspondence: zjh_mit@126.com

**Abstract:** Tea is one of the most popular drinks in the world. The rapid and accurate recognition of tea diseases is of great significance for taking targeted preventive measures. In this paper, an information entropy masked vision transformation (IEM-ViT) model was proposed for the rapid and accurate recognition of tea diseases. The information entropy weighting (IEW) method was used to calculate the IE of each segment of the image, so that the model could learn the maximum amount of knowledge and information more quickly and accurately. An asymmetric encoder–decoder architecture was used in the masked autoencoder (MAE), where the encoder operated on only a subset of visible patches and the decoder recovered the labeled masked patches, reconstructing the missing pixels for parameter sharing and data augmentation. The experimental results showed that the proposed IEM-ViT had an accuracy of 93.78% for recognizing the seven types of tea diseases. In comparison to the currently common image recognition algorithms including the ResNet18, VGG16, and VGG19, the recognition accuracy was improved by nearly 20%. Additionally, in comparison to the other six published tea disease recognition methods, the proposed IEM-ViT model could recognize more types of tea diseases and the accuracy was improved simultaneously.

**Keywords:** information entropy; masked autoencoder; vision transformer; tea disease image recognition

## 1. Introduction

In recent years, deep learning has been widely used in the diagnosis of crop diseases [1] and the recognition of biological genes [2]. Applying artificial intelligence (AI) methods to the diagnosis of crop diseases can provide a novel solution for agricultural safety production and green sustainable development, which is of great significance for ensuring the healthy growth of crops.

Tea is a green and healthy drink, very popular in China, with a drinking history of thousands of years, and is more and more popular with people all over the world. With the increasing scale and yield of tea planting year by year, tea diseases have seriously affected the quality, yield, and nutritional value. The diagnosis of tea diseases is a very challenging task. At present, the recognition of tea diseases is generally done manually. The workers not only need professional knowledge of tea diseases, but also need to have rich work experience. However, due to the limited number of tea pathologists and professional tea farmers, the diagnosis of tea diseases cannot be completed in a timely and effective manner. By using the AI methods instead of manually completing the recognition of tea diseases, the large-scale recognition and prevention of tea diseases can be realized more accurately and efficiently, thereby effectively reducing the economic losses caused by tea diseases.

At present, researchers have achieved many results of tea disease recognition based on AI. As early as 2015, Singh et al. [3] proposed an algorithm for the image segmen-

tation technique used for automatic detection and classification of plant leaf diseases. Hossain et al. [4] used the Support Vector Machine (SVM) classifier to recognize the tea leaf's diseases and the eleven features were analyzed during the classification. These features were then used to find the most suitable match for the disease (or normality) every time an image was uploaded into the SVM database. When a new image was uploaded into the system, the most suitable match was found and the disease was recognized. Chen et al. [5] used a CNNs model named Leaf-Net with different sized feature extractor filters that automatically extracted the features of tea plant diseases from images. DSIFT (dense scale-invariant feature transform) features were also extracted and used to construct a bag of visual words (BOVW) model that was then used to classify diseases via SVM and multi-layer perceptron (MLP) classifiers. Hu et al. [6] proposed a deep learning method to improve the performance of detection and severity analysis of tea leaf blight. A retinex algorithm was utilized to enhance the original images and reduce the influence of light variation and shadow. The leaves were detected using a deep learning framework named Faster Region-based Convolutional Neural Networks (FRCNN) to improve the detection performance of blurred, occluded, and small pieces of diseased leaves.

With the development of computer technology, methods such as image processing and machine learning have been widely used in crop disease recognition. The deep neural network is the basis of deep learning. The deep neural network has extraordinary performance in image recognition [7], classification, and detection, and has promoted the development of computer vision in the past ten years. In recent years, the field of natural language processing (NLP) has achieved unprecedented development with the introduction of the transformer and attention [8]. Although the transformer architecture has become the main method standard for NLP tasks, its application in computer vision is still limited. In computer vision, attention is either used in combination with the convolutional neural network (CNN) or used to replace some components of the convolutional network. This is because language is a signal with high semantics and dense information generated in human communication, while image is a natural signal with high spatial redundancy. With the introduction of the Vision Transformer (ViT) [9] model, the architectural gap has been resolved, and pure transformers directly applied to image patch sequences can perform image classification tasks well, while requiring very few resources for training computations. The emergence of ViT [10] transplanted from the language models in the computer vision community replaced the dominance of CNN [11]. In 2021, the masked autoencoders (MAE) were proposed [12], and it was proved that MAE are scalable self-supervised learners for computer vision. MAE developed an asymmetric encoder–decoder architecture. The encoder operates only on a subset of visible patches, and the decoder is only used during pre-training to recover the labeled masked patches and reconstruct missing pixels. Therefore, when combining these two designs to efficiently train larger models, the training speed can be improved by a factor of three or more.

Therefore, considering the existence of a large number of invalid background regions and redundant information in tea disease image samples, and the issues including poor resolution, image distortion, missing leaf images, and uneven image quality in a same tea disease image, the information entropy masked autoencoder-vision transformer (IEM-ViT) method has been proposed in this paper to further improve the accuracy of tea disease image recognition. By extracting the image information entropy and modeling the masking image according to the size of the information entropy, the input of redundant information and invalid background have been reduced, the model training speed has been sped up, and the recognition accuracy has been improved. At the same time, the tea diseases can be identified rapidly and accurately even under low image quality.

## 2. Materials and Methods

### 2.1. IE of Tea Disease Images

The backgrounds of tea disease images are usually complex, and there are many useless areas, which seriously affects the recognition accuracy. Removing the complex

background in the image and segmenting the diseased points can reduce the computational complexity and improve the speed and accuracy of recognition. In addition, the various color information of each region in the tea disease image has different contributions to the recognition of the tea disease. In this paper, the differences in the amount of information contained in these regional features are mainly described through the IE, so as to obtain the importance of each regional feature in the image. The IE of each segment of the image was firstly calculated before training, and the IEW method was used to enable the model to extract the effective features faster and more accurately.

IE [13] was firstly proposed to measure uncertainty and solve the problem of information quantification. As an objective weighting method, IE can avoid errors caused by human factors [14]. IE is defined in the information theory as the mathematical expectation of the random variable $I(x)$ in the set $\{X, q(x)\}$, and its mathematical expression is shown in Formula (1).

$$H(x) = -\sum_{x \in X} q(x) \log q(x) \tag{1}$$

In Formula (1), $H(x)$ represents the IE of $X$, and $q(x)$ represents the probability of occurrence of $X$. The larger the value of $H(x)$, the greater the uncertainty of $X$. When the random variable $x$ is a fixed value, its entropy is 0, and when $x$ obeys a uniform distribution, its entropy value is the largest. In a grayscale image, each pixel can be regarded as an independent variable $m$ (valued from 0 to 255), and the pixel points of the entire image can be regarded as a set $(m, p(m))$, where $p(m)$ represents the probability density of the occurrence of a point with a grayscale value of $m$. Then, according to the definition of IE in Formula (1), the representation of image IE $F$ can be obtained as shown in Formula (2).

$$F = \sum_{m=0}^{k} p(m) \log (p(m)) \tag{2}$$

In Formula (2), $k$ represents the gray value of the pixel ($k = 255$), and $p(m)$ represents the probability density of the pixel $m$ in the whole image. The image IE discussed in Formula (2) refers to the global IE, which represents the statistical distribution of all pixels in the whole image. Since the spatial distribution characteristics of image pixels are not considered, different images with the same probability distribution will have the same IE. In order to reasonably utilize the IE information of the image, the concept of unit IE is adopted in this paper [15]. Firstly, the grid descriptor is introduced. Figure 1 shows the grid description with a size of 7. As can be seen, by mapping the original tea disease image to the grid, a 7 × 7 units square matrix can be obtained. Based on the Formula (2), the global IE of each grid unit in Figure 1 can then be obtained, and the 7 × 7 IE matrix $E$ shown in Formula (3) is obtained.

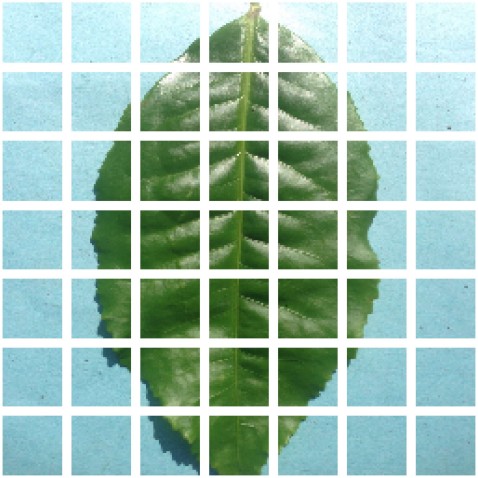

**Figure 1.** Grid description of size 7.

$$E = \begin{bmatrix} E_{11} & \dots & E_{1j} \\ \vdots & \ddots & \vdots \\ E_{i1} & \dots & E_{ij} \end{bmatrix}, 1 \le i \le 7, 1 \le j \le 7 \tag{3}$$

By introducing grid description and solving the unit entropy, the global IE matrix is obtained, which quantifies the importance of different regional features of the tea disease images. This makes the input features have a larger amount of information, and ultimately improves the recognition accuracy and stability of the model.

### 2.2. Tea Leaf's Disease Recognition

Deep learning methods are currently popular target recognition methods, but there is an overfitting problem when the training set is small. CNN models, such as ResNet, VGGNet, AlexNet, GoogLeNet, Faster R-CNN, and YOLO, have been applied to plant disease detection. Barbedo et al. using deep learning explores the plant disease recognition from individual lesions and spots [16]. Mohanty et al. analyzed the performance of AlexNet and GoogLeNet for image-based plant disease detection [17]. Liu et al. discussed the visual discrimination methods of citrus HLB based on features of images combined with hyperspectral imaging technology [18]. Ozguven et al. used a Faster R-CNN model to estimate the severity of diseases in sugar beet leaves [19]. Chen et al. used deep transfer learning for image-based plant disease recognition [20]. Wang et al. compared the performance of a series of deep CNN to estimate the severity of the apple black rot [21]. The research on the recognition of tea diseases mainly needs to consider the invalid and complex background of tea disease images, and remove redundant invalid information to speed up the training speed and improve the recognition accuracy. This paper uses the improved MAE to mask 75% of the invalid or unimportant areas according to the IE of tea disease images, and then uses the generated tea disease characteristic images to realize the sharing of training parameters to enhance the training samples and complete the accurate recognition of tea diseases.

#### 2.2.1. Vision Transformer

The rise of the transformer is mainly due to its good performance in the field of NLP. The attention mechanism solves the shortcomings of the recurrent neural network model, such as long short-term memory (LSTM) [22], it cannot be trained in parallel, and it requires a lot of storage resources to memorize the entire sequence of information [23]. The transformer uses an acyclic network structure and performs parallel computing through the encoder–decoder and self-attention mechanism, which greatly shortens the training time and improves the performance of machine translation [24]. Based on the transformer model [25–27], bidirectional encoder representations from transformers (BERT) [28] is designed for pretrain deep bidirectional representations from unlabeled text by jointly conditioning on both left and right context in all layers. As a result, the pre-trained BERT model can be finetuned with just one additional output layer to create state-of-the-art models for a wide range of tasks, such as question answering and language inference, without substantial task specific architecture modifications. The transformer is applied to computer vision to build a global information interaction mechanism, which helps to establish a more adequate feature representation. In addition, ViT also adopts the standard data flow form in the transformer, which facilitates efficient fusion with other modal data. The transformer-based vision model performs better than the convolutional neural network in image classification [29], object detection [30], image segmentation [31], video semantic understanding [32], image generation [33], and other fields. Figure 2 shows the model structure of the proposed ViT for tea diseases.

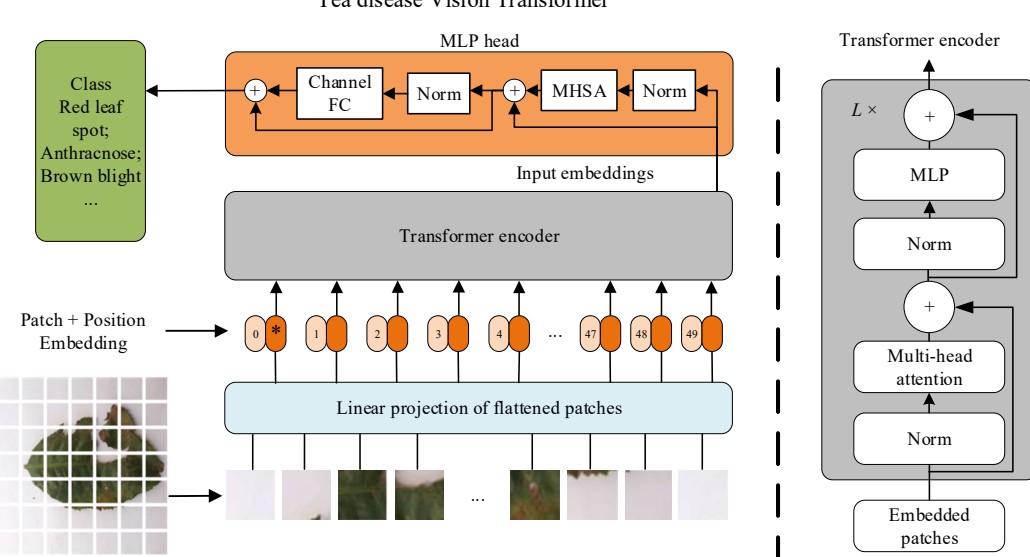

**Figure 2.** Model structure of Vision Transformer (ViT) for tea diseases.

### 2.2.2. Self-Supervised Learning

In the task of crop disease image recognition, there is usually a large number of unlabeled samples. Accurate and meticulous labeling of crop disease images requires not only professional agricultural pathology knowledge but also certain computer application technology. Labeling is difficult and a lot of labor is needed.

Self-supervised learning is a form of unsupervised learning where the data itself provides a strong supervisory signal that enables CNN to capture the intricate dependencies in data but without the need for external labels. Self-supervised learning allows the learning of generalized and semantic representation features from unlabeled data without relying on huge, labeled datasets by building deep learning models whose performance matches the supervised objects.

In this paper, the method based on self-supervised learning uses the ImageNet data set for pre-training to obtain the initial model parameters, and then transfers the parameters to the tea disease recognition task, and finally uses the ViT model to fine-tune the parameters for tea disease recognition. Since the data in the pre-training stage does not need to be labeled, the cost of labeling data is greatly reduced to improve the recognition efficiency and accuracy of the target tasks [34].

### 2.2.3. Masked Visual Autoencoder

Masked language modeling, such as BERT and GPT [35], are very successful pre-training methods in NLP. This method has better scalability and performance in various downstream tasks such as image [36], video [37], and multimodality [38]. Autoencoders are a class of neural networks used in unsupervised learning, and their function is to perform representation learning on the input information as a learning target. Autoencoder is a type of neural network used in semi-supervised learning and unsupervised learning. Its function is to perform representation learning by taking the input information as a learning target. It has the function of a general representation learning algorithm and can be applied to computer vision problems, including image noise reduction, image style transfer, and image completion. Denoising autoencoders (DAE) [39] are a class of autoencoders that corrupt an input signal and learn to reconstruct the original, uncorrupted signal. Due to the complex background of tea disease images, there are useless areas, which affect the recognition accuracy. In order to remove the complex background in the image and segment the diseased points, the method of denoising the automatic coding is used in this paper to build an autoencoder.

According to the ViT model, the tea disease images are divided into regular non-overlapping patches, then a subset of patches are sampled while the remaining patches are masked. Through random sampling with a high masked rate (75% masked), the redundancy of tea disease image information is largely eliminated. At the same time, the possible center deviation is avoided by uniformly distributed sampling, which prevents too much masking of the patches near the center area of the tea disease image during masking. Through the random mask sampling of tea disease images, highly sparse image information features are obtained, which reduces the redundant information of input images and makes the operation of the autoencoder more efficient.

In the standard ViT model, the encoder embeds patches through linear projection and adds positional embeddings. However, in this paper the encoder operated on only a small fraction (25% of images) of the full set of tea leaf disease images. As a result, it was possible to train the larger datasets with only a small fraction of the memory, while greatly reducing the computational load of the encoder. The input to the decoder was the full set of tokens consisting of the encoded visible patches and mask tokens. Each mask token was a shared learned vector that represented the location information of the image patch in the full image. The decoder was used during pre-training to perform image reconstruction tasks.

In this paper, an asymmetric encoder–decoder structure was used, and the structure of the decoder was smaller than that of the encoder, which can significantly reduce the pre-training time. The MAE structure of the tea disease images is shown in Figure 3.

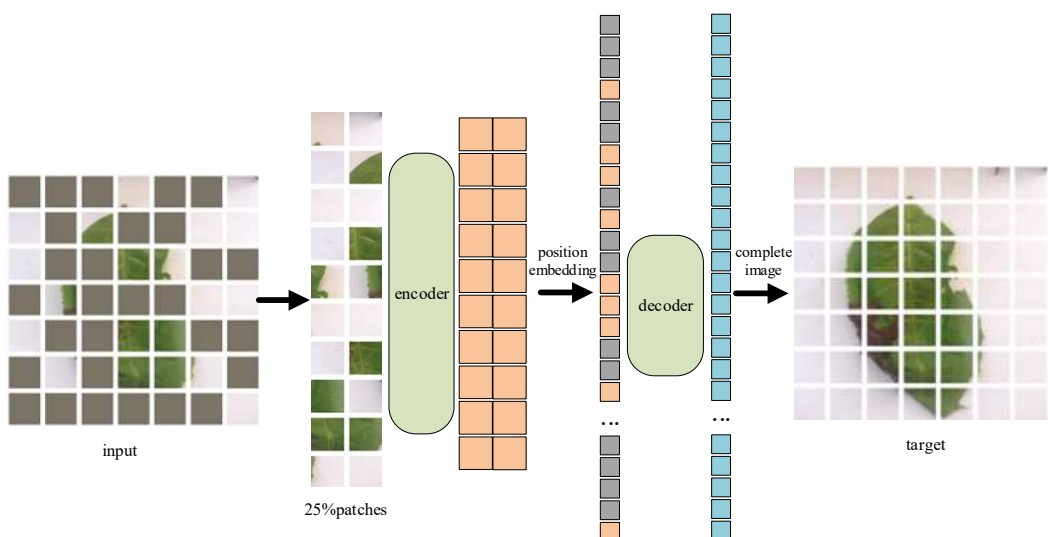

**Figure 3.** Masked autoencoder (MAE) structure of tea disease images.

## 2.3. Dataset Description

The tea disease images dataset was collected in Johnstone Boiyon farm, Koiwa location, Bomet county. This dataset contained seven common tea leaves disease images: (1) Red leaf spot, (2) Algal leaf spot, (3) Bird's eye spot, (4) Gray blight, (5) White spot, (6) Anthracnose, and (7) Brown blight. There were a total of 885 tea disease images and each of the classes contained more than 100 images. In addition, the dataset also contained a class of healthy tea leaves. The sampled tea disease images of each class are shown in Figure 4. It can be seen from Figure 4 that the image samples of the same disease have uneven image quality, including resolution, color depth, image distortion, leaf breakage, and other aspects. In this paper, the method of IEW enabled the model to learn the knowledge with the maximum amount of information more quickly and accurately, so that the model could extract the more effective features.

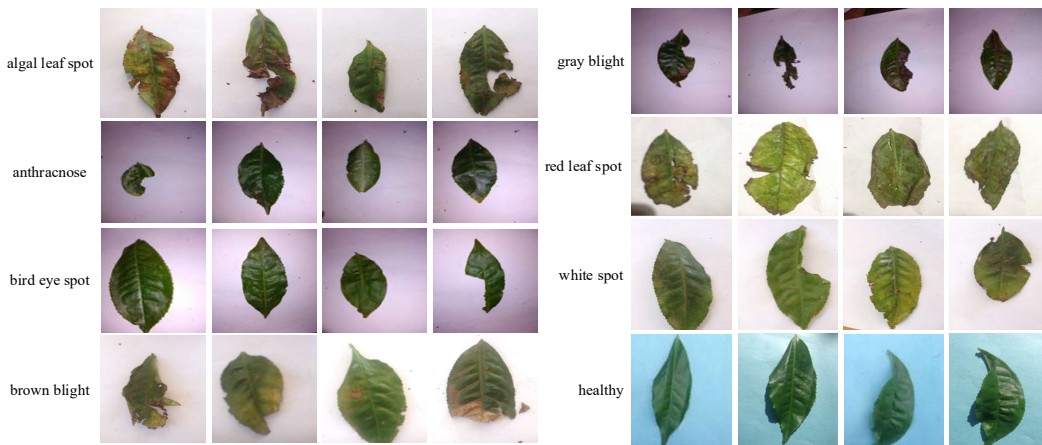

**Figure 4.** Types of tea disease images.

As shown in Figure 5, seven kinds of tea diseases and healthy image data are firstly divided into the training data set and test data set. The training data set accounts for 80% (including 708 pictures), and the test data set accounts for 20% (including 177 pictures).

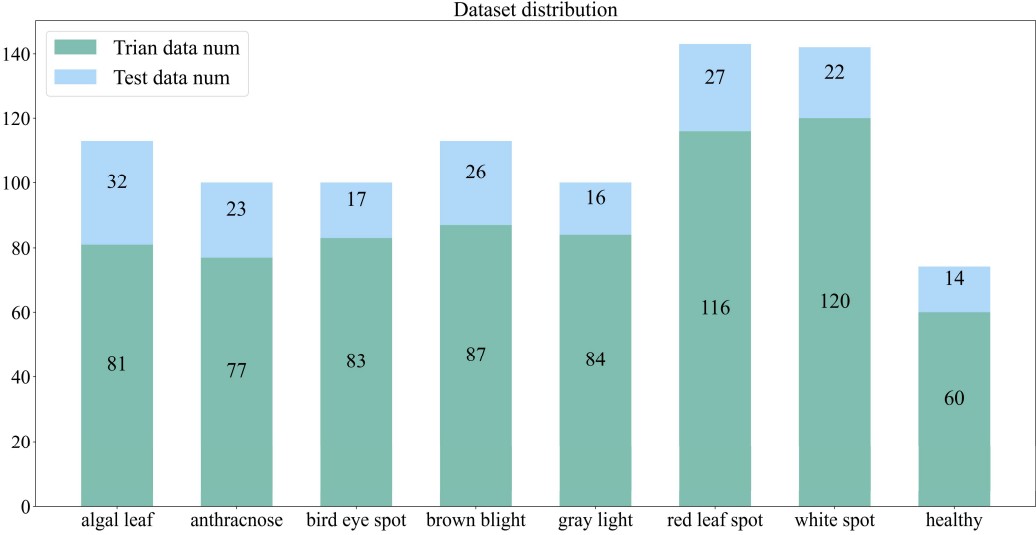

**Figure 5.** Distribution of the tea disease dataset.

### 2.4. Tea Disease Recognition

Figure 6 shows the entire process of model training and testing. Two steps, including the feature extraction training and classifier training, were performed on the training set. In the feature extraction step, an IEW-based MAE was used to complete the masked tea disease image to obtain a complete tea disease image. Then, the information features and model parameters enhanced by the MAE were extracted and shared with the classifier. In the classification model training step, the parameters and informative features of the image completion model were loaded, and then the discriminator model was connected for complete model training. Finally, the trained model was verified to use the test set and obtain the model accuracy.

In the first stage of the ViT model training, an IE auto-encoding model was designed. We set the input $x \in \{0, 1\ 2, \dots, 255\}$ and performed a fragmentation process on $x$, $p = patch(X)$, then converted each fragment into a grayscale image and calculated the IE of each fragment $N = normal[entropy(p)]$, finally, the randomly generated mask matrix $M$ was weighted to obtain a new mask matrix $M'$. The input data was obtained through matrix transformation. The transformation module is shown in Formula (4).

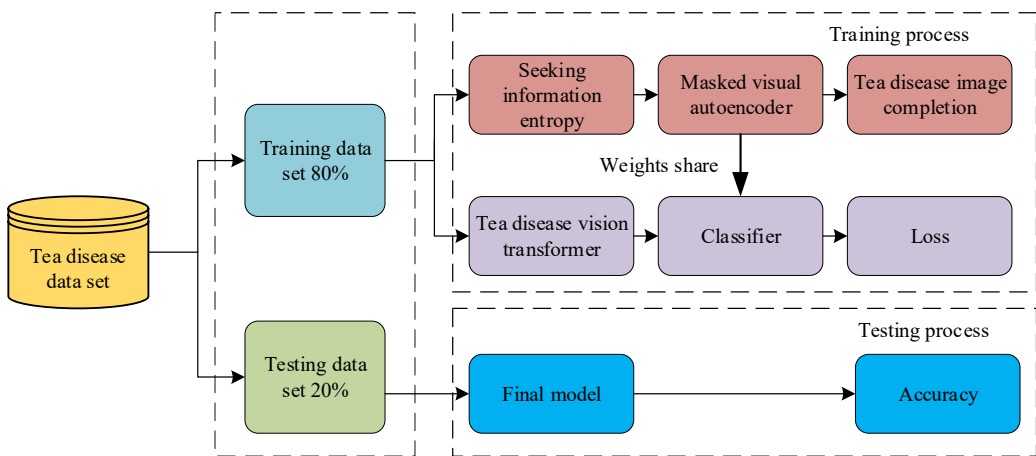

**Figure 6.** Process of the tea disease recognition model.

$$
\begin{aligned}
P \quad &= M'p \\
&=
\begin{bmatrix}
p_{(0,0)} w_{(0,0)} & p_{(0,1)} w_{(0,1)} & \cdots & p_{(0,n)} w_{(0,n)} \\
p_{(1,0)} w_{(1,0)} & p_{(1,1)} w_{(1,1)} & \cdots & p_{(1,n)} w_{(1,n)} \\
\vdots & \vdots & \ddots & \vdots \\
p_{(m,0)} w_{(m,0)} & p_{(m,1)} w_{(m,1)} & \cdots & p_{(m,n)} w_{(m,n)}
\end{bmatrix}
\end{aligned}
\tag{4}
$$

In Formula (4), $p_{i,j}$ are the slices of the input image, and $w_{i,j}$ are the weights obtained by calculating the IE of the slices. By changing values of $w_{i,j}$, the attention of the network can be effectively adjusted, and the learning attention of the network information can be transferred to the outline of the tea disease, rather than the background of the image or other invalid areas. As a result, the network can better learn more important information in the image, so as to adapt a variety of complex tea diseases and improve the accuracy of the network. The specific algorithm of the IEW module of tea disease images is shown in Algorithm 1.

---

**Algorithm 1:** Image Fragmentation IE Weighting Algorithm

---

Input patches (R G B), mask_ratio
Output: input autoEncode model patches
1.  Entr = {0,0, . . . ,0}
2.  L = len(patch)
3.  for patch →patches:
4.      I_R(x,y), I_G(x,y),I_B(x,y) = patch(R G B)
5.      img(x,y) = 1/3*I_R(x,y) + 1/3*I_G(x,y) + 1/3*I_B(x,y)
6.      hist = histogram(img, bins = range(0, 256))
7.      hist = hist[hist > 0]
8.      res = −log2(hist/hist.sum()).sum()
9.      entr[i] = res
10. len_keep = int(L * (1 − mask_ratio))
11. ids = argsort(random(L)* entr)[0:len_keep]
12. return patch[ids]

---

In Algorithm 1, the input items were the segmented image blocks $p_{i,j}$ and the proportion (75%) that needed to be masked in the training process, and the output item was the image block matrix $P$ of the input MAE model. The 1st and 2nd lines in the algorithm were used to initialize the IE vector and the total amount of image blocks, the 3rd line was used to calculate the IE of the image, the 4th and 5th lines were used to convert the RGB image into a grayscale image, the 6th, 7th, and 8th lines were used to calculate the IE of the grayscale image and save it into the IE vector, the 10th line was used to calculate the

number of image blocks to be reserved, and the 11th line was used to randomly sample the image blocks and select the image blocks by weighting according to the IE weight obtained in the 3rd line.

When performing image masking, the weighted size $w_{i,j}$ of the IE was used to select suitable shards for masking, and retaining the shards with larger $w_{i,j}$ for feature extraction, which can prevent the model from overfitting. The encoder module of the feature extraction model contained a total of 8 transformer layers, and adopted a multi-head attention structure, respectively, expressed as matrix *query* (*Q*), *key* (*K*), *value* (*V*). Specifically, feature extraction was performed through the following three steps:

Step 1. Similarity calculating and comparing of matrix *Q* and matrix *K*. The similarity is represented by *f*

$$f(Q, K_i), i = 1, 2, \ldots, m \tag{5}$$

Step 2. Similarity normalizing through the SoftMax function to obtain the weight value $\alpha_i$

$$\alpha_i = \frac{e^{f(Q, K_i)}}{\sum_{j=1}^{m} f(Q, K_i)}, i = 1, 2, \ldots, m \tag{6}$$

Step 3. Weighted sum calculation on all the values in $V_i$ according to the calculated weight $\alpha_i$. The obtained feature vector *FV* is

$$FV = \sum_{i=1}^{m} \alpha_i V_i \tag{7}$$

The whole process can be expressed as

$$Attention(Q, K, V) = softmax(\frac{Q^T K}{\sqrt{d_k}})V \tag{8}$$

In Formula (8), $Q \in R^{m \times dk}$, $K \in R^{m \times dk}$, $V \in R^{m \times dv}$, and the output dimension is $R^{m \times dv}$. In the decoding stage, the transformer network module was also used.

In the classification stage of the IEM-ViT network, the encoder model parameters learned in the first stage were shared with the classifier recognition model to recognize the tea diseases. The classifier model can be expressed as

$$Classes = softmax[FC(features)] \tag{9}$$

In Formula (9), features are the eigenvalues extracted by the encoder network model. The input of the classifier at this stage was the complete image of tea diseases obtained after completion. In order to reduce the computational load of the network model, the image embedding coding was necessary before inputting the model, and then extracting the information in the tea image through the ViT module. Finally, the classifier discriminator was used to recognize the tea disease and cross entropy was used to calculate the loss.

## 3. Experimental Results and Analysis

Considering CNNs are one of the current popular image recognition methods, in these experiments, several main CNNs models were performed and compared with the proposed IEM-ViT model in the tea disease recognition.

The VGG model was proposed in 2014 to extract image features through a deeper network structure and small convolution kernels, including two main models, the VGG16 and VGG19. The VGG16 contains 16 convolutional layers to extract features while the VGG19 contains 19 convolutional layers to extract features, and then to realize image recognition through the fully connected layers. The ResNet model adopts the residual network module to make the network structure deeper, and it has also achieved remarkable results in image processing.

### 3.1. Model Parameters

Deep learning often requires a large number of samples in the training process to achieve good results, but due to the small sample size of the dataset used in this paper, we choose to train the model by loading the parameters trained by the ImageNet dataset. Some parameter settings of the proposed IEM-ViT model are shown in Table 1.

**Table 1.** Model parameters.

| Parameter | Value | Description |
|---|---|---|
| Target_size | $224 \times 224 \times 3$ | Size of the image we fed into the model |
| Batch_size | 16 | Number of images in a batch |
| Mask_ratio | 0.75 | Masking ratio |
| Patch_size | $16 \times 16 \times 3$ | Size of the image patch |
| Number_heads | 16 | Number of transformer heads |

### 3.2. Tea Disease Image Completion

Slices with a resolution of $16 \times 16$ were masked during the autoencoder processing stage. The random method was used to mask 75% of the patches on the basis of the IEW. The masked patches were occupied by a 0 matrix, and only 25% of the patches in the image input to the model had the original data information. After training, the model could complete the masked missing information, so as to obtain the feature parameters required for the tea disease recognition. By randomly masking different parts of each image many times, the scale of the training data set was effectively increased, which resulted in the problem of overfitting being avoided, and the goal of learning more features for the model being achieved. The completion result is shown in Figure 7. It can be seen from Figure 7 that the input masked tea image data can be well completed, indicating that although only 25% of the original image data was input, the proposed IEM-ViT model could obtain enough tea disease feature information.

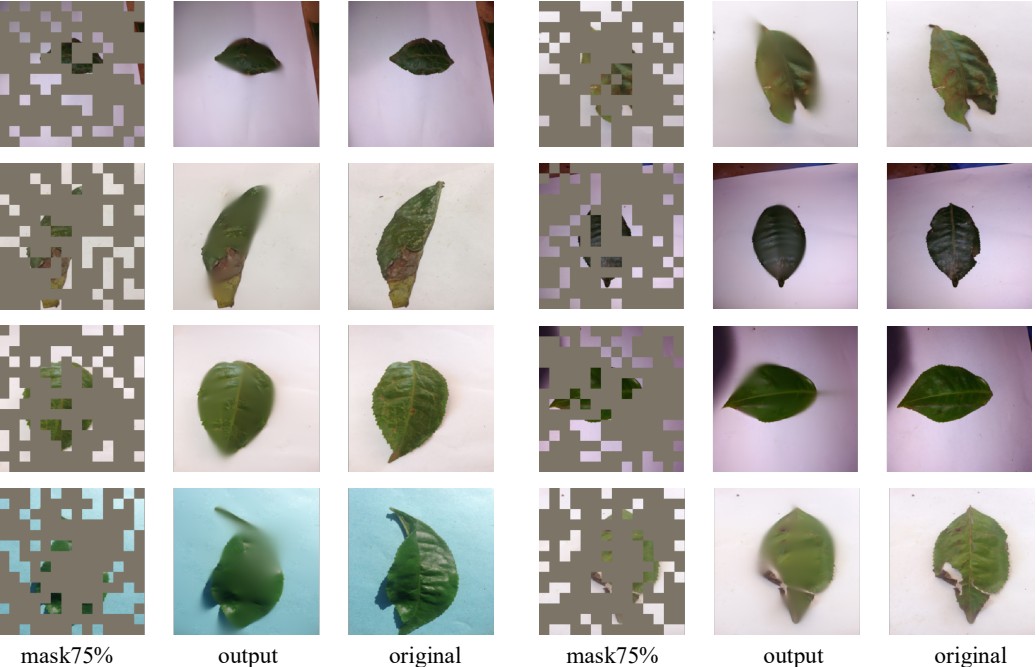

**Figure 7.** Completion results of the tea disease images.

### 3.3. Experimental Results

The proposed IEM-ViT model was compared with three current popular deep learning classification models including the VGG16, VGG19, and ResNet18. In order to ensure the

validity of the experimental comparison, all models were trained with the same learning rate. The loss, top1 accuracy, and top5 accuracy of each model in the last 50 epochs during the training process were calculated, respectively. The results are shown in Figure 8. As can be seen from Figure 8, when comparing with the three classification models of VGG16, VGG19, and ResNet18, the proposed IEM-ViT model was more stable in terms of training loss and testing loss, and the loss value after model convergence was also smaller. In addition, the proposed IEM-ViT model had a top one accuracy of 93.78% and a top five accuracy of 100%.

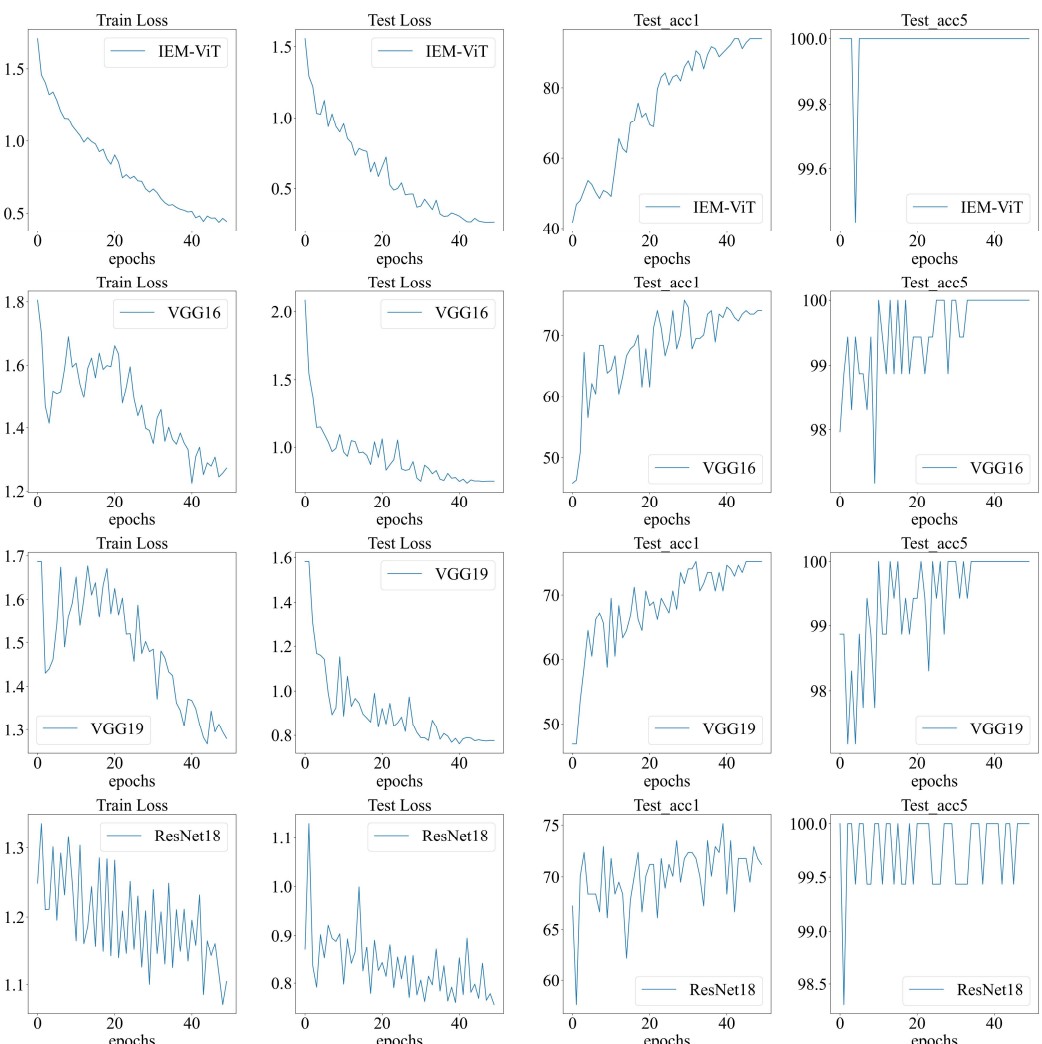

**Figure 8.** Loss and accuracy of each model.

Table 2 shows the comparison between the proposed IEM-ViT model and the other three models on the four evaluation indicators of accuracy, precision, recall, and F1-score. As seen from Table 2, the IEM-ViT model accuracy was 93.78% with an improvement of 18.68% (93.78–75.10) compared with the best performing model (ResNet18) among the other three models, and an improvement of 22.03% (93.78–71.75) compared with the worst one (VGG19). In terms of precision, the IEM-ViT model was 0.9367, which was an improvement of 0.1517 (0.9367–0.7850) compared with the VGG16, and 0.1886 (0.9367–0.7481) compared with the VGG19. In terms of recall, the IEM-ViT model was 0.938, which was improved by 0.1632 (0.9380–0.7748) compared with the ResNet18. Additionally, the F1-score was improved by 0.1744 (0.9364–0.7620) compared with the ResNet18. As a result, it can be seen that the proposed IEM-ViT model outperformed the other three models in the four evaluation indicators of accuracy, precision, recall, and F1-score.

**Table 2.** Tea disease recognition results of different models.

| Model | Accuracy (%) | Precision | Recall | Fi-Score |
|---|---|---|---|---|
| IEM-ViT (this work) | 93.78 | 0.9367 | 0.9380 | 0.9364 |
| VGG16 | 73.40 | 0.7850 | 0.7694 | 0.7557 |
| VGG19 | 71.75 | 0.7481 | 0.7429 | 0.7336 |
| ResNet18 | 75.10 | 0.7750 | 0.7748 | 0.7620 |

In addition, Figure 9 shows the confusion matrix for the tea disease recognition. The recognition performance of the model for the seven tea disease types and the healthy model can be fully seen from Figure 9. The values on the diagonal in Figure 9 are the number of correctly classified samples, while the values not on the diagonal are the number of misclassified samples.

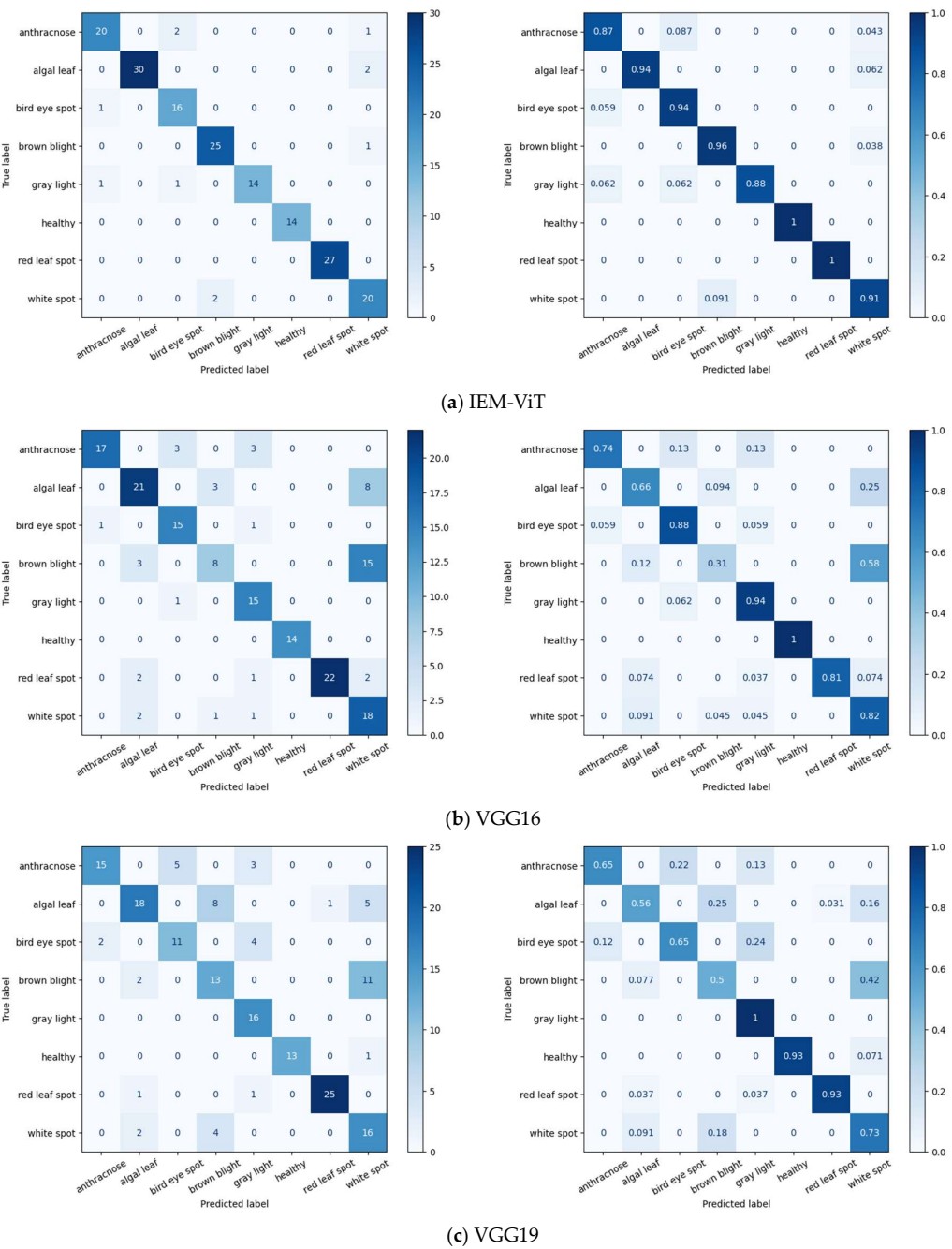

(**a**) IEM-ViT

(**b**) VGG16

(**c**) VGG19

**Figure 9.** *Cont*.

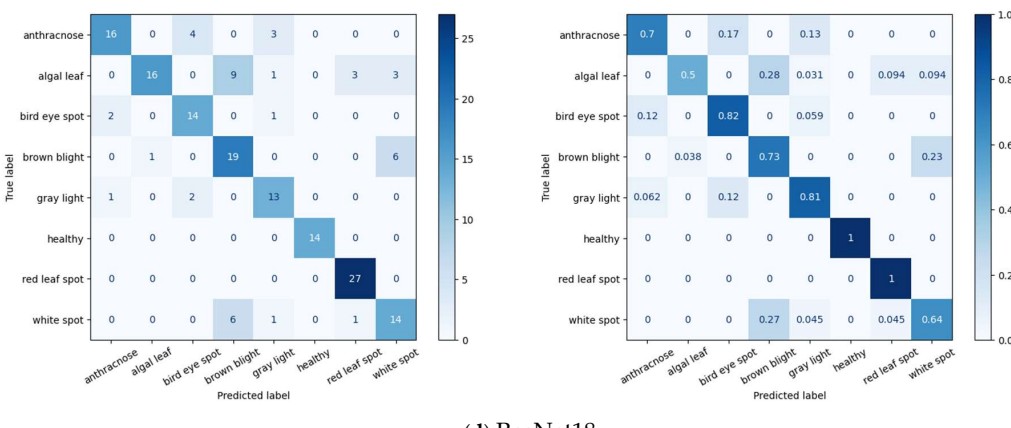

(**d**) ResNet18

**Figure 9.** Tea disease recognition confusion matrix.

As shown in Figure 9a, in the test set, two samples of anthracnose were incorrectly identified as bird eye spots, and one sample was incorrectly identified as white spots. The recognition accuracy of the anthracnose class was 87%, which was the worst performance. Two samples in the algal leaf were incorrectly identified as white spots, with an accuracy of 94%. One sample of bird eye spot was incorrectly identified as anthracnose, with an accuracy of 94%. One sample of brown light was incorrectly identified as a white spot, with an accuracy of 96%. One sample in gray light was incorrectly identified as anthracnose and the other as bird eye spot, with an accuracy of 88%. Two samples in the white spot were incorrectly identified as brown light, with an accuracy of 91%.

As shown in Figure 9b, for the VGG16 model, three samples of brown light were incorrectly identified as algal leaf and fifteen samples were incorrectly identified as white spot, with an accuracy rate of only 31%. This kind of tea disease has the worst recognition accuracy. As shown in Figure 9c, for the VGG19, two samples of brown light are incorrectly identified as algal leaf, and eleven samples were incorrectly identified as white spot, with a recognition accuracy of only 50%. As shown in Figure 9d, for the ResNet18, nine samples in the algal leaf were incorrectly identified as brown light, one sample was incorrectly identified as gray light, three samples were incorrectly identified as red leaf spot, three samples were incorrectly identified as white spot, and the recognition accuracy was only 50%. From the tea disease recognition confusion matrix in Figure 9, it can be seen that algal leaf and brown blight tea diseases were the most difficult to distinguish in the model.

Furthermore, Table 3 gives the comparison of other published tea disease recognition algorithms and the proposed IEM-ViT model in terms of disease type recognition and accuracy. As can be seen from Table 3, in comparison to the LeafNet method, for the same types of tea diseases, the recognition accuracy was improved by 3.62% (93.78–90.16%). In comparison to the other five methods, the proposed IEM-ViT model could recognize more types of tea diseases and the recognition accuracy was improved simultaneously.

**Table 3.** Comparison of the proposed IEM-ViT and other methods.

| Model | Reference | Types Evaluated | Accuracy (%) |
| --- | --- | --- | --- |
| Concatenated CNN | Krisnandi et al. [40] | 4 | 89.64 |
| VGG16 | Hu et al. [41] | 3 | 90 |
| LeafNet | Chen et al. [5] | 7 | 90.16 |
| TLDR | Karmokar et al. [42] | 1 | 91 |
| SVM | Hossain et al. [4] | 2 | 91 |
| Improved deep CNN | Hu et al. [43] | 4 | 92.5 |
| IEM-ViT | This paper | 7 | 93.78 |

## 4. Conclusions

The tea leaf diseases have seriously affected the yield and quality of tea. At present, the recognition of these diseases mainly relies on agricultural pathologists, resulting in lower recognition efficiency and the inability to take timely and effective targeted prevention measures. In this paper, the IEM-ViT model was proposed to realize the tea image diseases recognition with an improved recognition accuracy.

Since there are many useless areas in the tea disease image, removing redundant information in the image and segmenting the disease points can improve the recognition accuracy, reduce the computational complexity, and speed up the recognition speed. This paper mainly described the difference in the amount of information contained in the regional features through the IE, so as to obtain the importance of each regional feature in the image. The IE was calculated for each segment of the image before training, and the IEW method enabled the model to learn the knowledge with the largest amount of information faster and more accurately. The labeled, masked image patches were then complemented by a MAE to reconstruct the missing pixels for parameter sharing and data augmentation. Therefore, in the case of missing images or low image quality, such as low resolution, it can also have a higher recognition accuracy.

The experimental results showed that the recognition accuracy of the proposed IEM-ViT model for the seven types of tea diseases, including the algal leaf, bird eye spot, brown light, gray light, red leaf spot, and white spot, reached 87%, 94%, 94%, 96%, 88%, 100%, and 91%, respectively. The average accuracy rate was 93.78%, which is a nearly 20% improvement in comparison to the three currently main image recognition algorithms, including the ResNet18, VGG16, and VGG19. Additionally, in comparison to the published tea diseases method LeafNet, for the same types of tea diseases, the recognition accuracy is improved by 3.62%, and in comparison to the other five methods, our method can recognize more types of tea diseases and the accuracy is improved simultaneously.

In future research, according to the characteristics of different crop diseases, the proposed method can also be extended to the recognition of other crop diseases, so as to provide a new method for taking scientific and precise prevention measures and realizing the sustainable development of crop production.

**Author Contributions:** Conceptualization, H.G. and J.Z. (Jiahong Zhang); methodology, J.Z. (Jiahong Zhang); software, H.G.; validation, J.Z. (Jiahong Zhang), H.G. and J.Z. (Jing Zhang); formal analysis, J.Z. (Jiahong Zhang); investigation, H.G.; resources, J.Z. (Jiahong Zhang); data curation, J.G.; writing—original draft preparation, H.G.; writing—review and editing, J.Z. (Jiahong Zhang); visualization, J.G.; supervision, J.Z. (Jiahong Zhang); project administration, J.Z. (Jing Zhang). All authors have read and agreed to the published version of the manuscript.

**Funding:** This work was supported by the National Natural Science Foundation of China (62162034), the basic research program general project of Yunnan province (202201AT070189) and the basic research program key project of Yunnan province [NO. 202101AS070016].

**Data Availability Statement:** The data we use comes from the public dataset on the Kaggle website, https://www.kaggle.com/code/rizqyad/tea-leaves-disease-classification/data. (accessed on 14 April 2023).

**Conflicts of Interest:** The authors declare no conflict of interest.

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
