# Peer review of "An Information Entropy Masked Vision Transformer (IEM-ViT) Model for Recognition of Tea Diseases"

_agronomy, doi:10.3390/agronomy13041156_

Round 1

Reviewer 1 Report

In the presented paper, the authors apply the information entropy weighting method to improve the deep learning technique for detecting tea diseases. The recognition accuracy of the proposed method is around 20% higher than that of the currently existing known detection methods. Results are new and of high relevance and the paper deserves being published.

Nevertheless, I would appreciate if the authors could decribe more expicitely the comparison of the amount of the processed data / working time / impact on the accuracy of the deep learning method applied later when using information entropy preprocessing method and without using it. 

Author Response

Dear reviewer:

We are very grateful to your comments for the manuscript. According to your advices, we have revised the relevant parts in manuscript. Some of your questions were answered below.

According to your suggestion, the authors think that the amount of the processed data may have impacts on the accuracy of the deep learning method applied later using information entropy preprocessing method. However, due to the limited time how and what extent the impact will occur will be considered in the future work.

Reviewer 2 Report

The paper 'An information entropy masked vision transformer (IEM-ViT) model for recognition of tea diseases’ provides an interesting and innovative study. I do not have any major revisions for the paper, so my recommendation was for 'Accept after minor revision'.

 Additional comments 

1) The research objectives must be clear in the last paragraph of the Introduction. It is not common to present the main contributions of the paper in the Introduction section. Review that section (lines 74-86).

 2) Section 2 related work is not common in a paper. Topic 2 could be inserted in Introduction and Material and Methods. Please readjust the texts of lines 112 to 259.

 3) Define all acronyms used in the titles of Tables and Figures. For example: Figure 3. Masked autoencoders (MAE) structure of tea disease images.

Author Response

Dear reviewer:

We are very grateful to your comments for the manuscript. According to your advices, we have revised the relevant parts in manuscript. Some of your questions were answered below.

  1. According to your suggestion,the research objectives have been shown in the last paragraph of the Introduction.Besides, the main contributions(lines 74-86 of the previous manuscript) in the introduction section have been deleted.
  2. According to your suggestion, part of related work in Section 2 (lines 88-111 in the previous manuscript) has been included in introduction section, while the other part of related work in section 2 (lines 112-259 in the previous manuscript) has been included in Material and Methods section.
  3. According to your suggestion, all the acronyms used in the titles of Tables and Figures have been defined.
